# MCL1 Inhibition Overcomes the Aggressiveness Features of Triple-Negative Breast Cancer MDA-MB-231 Cells

**DOI:** 10.3390/ijms241311149

**Published:** 2023-07-06

**Authors:** Giovanni Pratelli, Daniela Carlisi, Diana Di Liberto, Antonietta Notaro, Michela Giuliano, Antonella D’Anneo, Marianna Lauricella, Sonia Emanuele, Giuseppe Calvaruso, Anna De Blasio

**Affiliations:** 1Department of Physics and Chemistry (DiFC)-Emilio Segrè, University of Palermo, 90128 Palermo, Italy; giovanni.pratelli@unipa.it; 2Section of Biochemistry, Department of Biomedicine, Neurosciences and Advanced Diagnostics (BIND), University of Palermo, 90127 Palermo, Italy; daniela.carlisi@unipa.it (D.C.); diana.diliberto@unipa.it (D.D.L.); marianna.lauricella@unipa.it (M.L.); sonia.emanuele@unipa.it (S.E.); 3Laboratory of Biochemistry, Department of Biological, Chemical and Pharmaceutical Sciences and Technologies (STEBICEF), University of Palermo, 90127 Palermo, Italy; antonietta.notaro@unipa.it (A.N.); michela.giuliano@unipa.it (M.G.); antonella.danneo@unipa.it (A.D.); giuseppe.calvaruso@unipa.it (G.C.)

**Keywords:** TNBC, anoikis resistance, EMT, cancer stem cells, MCL1, BH3-mimetic

## Abstract

Triple-Negative Breast Cancer (TNBC) is a particularly aggressive subtype among breast cancers (BCs), characterized by anoikis resistance, high invasiveness, and metastatic potential as well as Epithelial–Mesenchymal Transition (EMT) and stemness features. In the last few years, our research focused on the function of MCL1, an antiapoptotic protein frequently deregulated in TNBC. Here, we demonstrate that MCL1 inhibition by A-1210477, a specific BH3-mimetic, promotes anoikis/apoptosis in the MDA-MB-231 cell line, as shown via an increase in proapoptotic markers and caspase activation. Our evidence also shows A-1210477 effects on Focal Adhesions (FAs) impairing the integrin trim and survival signaling pathways, such as FAK, AKT, ERK, NF-κB, and GSK3β-inducing anoikis, thus suggesting a putative role of MCL1 in regulation of FA dynamics. Interestingly, in accordance with these results, we observed a reduction in migratory and invasiveness capabilities as confirmed by a decrease in metalloproteinases (MMPs) levels following A-1210477 treatment. Moreover, MCL1 inhibition promotes a reduction in EMT characteristics as demonstrated by the downregulation of Vimentin, MUC1, DNMT1, and a surprising re-expression of E-Cadherin, suggesting a possible mesenchymal-like phenotype reversion. In addition, we also observed the downregulation of stemness makers such as *OCT3/4*, *SOX2*, *NANOG*, as well as CD133, EpCAM, and CD49f. Our findings support the idea that MCL1 inhibition in MDA-MB-231 could be crucial to reduce anoikis resistance, aggressiveness, and metastatic potential and to minimize EMT and stemness features that distinguish TNBC.

## 1. Introduction

Breast cancer (BC) is a highly heterogeneous group of diseases characterized by different clinical behaviors and diverse biological characteristics that make the process of prediction and oncologic management more difficult [1,2]. The molecular classification of BC is based on the expression of the estrogen receptor (ER), progesterone receptor (PR), and human epidermal growth factor receptor 2 (HER2), resulting in four main subtypes which are luminal A, luminal B, HER2-positive (HER2+), and triple-negative breast cancer (TNBC). Among all BC subtypes, luminal A and B account for 70%, HER2+ accounts for 15–20%, and TNBC accounts for 15–20%. Each BC subtype has specific molecular characteristics, clinical behavior, prognosis, and treatment modalities [3]. 

TNBC refers to breast neoplasms that do not express ER, PR, and HER2 receptors on their cell surface. As shown by the high relapse and low survival rates, with respect to hormone receptor or HER2-positive BC subtypes, TNBC has the most aggressive clinical outcome, with early onset, great metastatic potential, and poor clinical prognosis [4]. In recent years, different groups in TNBC subtype were identified: Basal-like-1, Basal-like-2, Immunomodulatory, Mesenchymal, Mesenchymal-Stem-Cell-like, and Luminal Androgen, depending upon expression of distinct genes [5,6]. Nowadays, there is a high and constant increase in the number of clinical trials targeting TNBC. Current studies are conducted to identify new treatment possibilities, especially targeting transcriptome dysregulation and the immune microenvironment of TNBC patients [6,7,8]. 

One of the main obstacles to metastasis is a type of programmed cell death known as “anoikis” that is initiated by the loss of cellular contacts from the native ECM or neighboring cells triggering apoptosis. This process prevents these cells, which have detached from primary tumors, from reaching, adhering to, and growing in other body organs. Upon detachment, anoikis is induced by canonical apoptotic pathways which result in caspase activation [9]. However, metastatic tumors, including TNBC, developed a variety of strategies to enhance invasiveness and anoikis resistance including up-regulation of survival pathways and inactivation of p53 E-Cadherin lose expression [10,11,12]. In this context, the destruction of cell-to-ECM adhesion as well as cell-to-cell contacts via E-Cadherin inactivation promote epithelial–mesenchymal transition (EMT). In addition, several cellular mechanisms, including EMT, are regulated through AKT-mediated NF-κB activation, which is associated with increased invasiveness and anoikis resistance [13].

The differentiation state of the cancer cell, dynamic genomic alterations, as well as the clonal evolution and selection are among the causes that lead to phenotypic and molecular diversity within the tumor [14,15]. EMT and its inverse program called mesenchymal–epithelial transition (MET) are two mechanisms supporting phenotypic plasticity and intratumoral heterogeneity. During EMT, cells can be subject to dynamic changeovers from an epithelial state to a mesenchymal phenotype with consequent increase on migratory and invasive properties [16]. In this contest, anomalous activation of EMT by cancer cells plays a critical role in tumor progression due the increase in their aggressive behavior and facilitating the diffusion of metastatic cancer cells. However, the cancer cells that reverse to epithelial phenotype involved in MET are able to colonize distant organs and form metastasis with higher efficiency. For this reason, both EMT and MET programs influence drug resistance, improve survival and promote stemness state of both normal and neoplastic mammary epithelial cells [17]. Nowadays, the mechanisms involved in EMT are increasingly being studied. Several studies highlight new insights into EMT/MET processes, especially relate genetic signature [18], epigenetic alterations implication [19,20], structural alterations of the extracellular matrix (ECM) [20] as well as the interplay between immune system [21,22] and metabolic reprogramming (MR) [22,23], and how these factors affect the tumor growth and progression to metastasis of TNBC.

Several pieces of evidence suggest that TNBC cells are enriched in cancer stem cells (CSCs), which markedly contribute to therapy resistance and failure, a phenomenon defined as multi-drug resistance (MDR), leading to recurrence and an increase in mortality. CSCs are a small population of self-renewing tumor cells able to persist after therapy and, thanks to their ability to differentiate into all the cell types within the original tumor, can reinstate their heterogeneity. The deregulation of stemness pathways is even more enhanced in TNBC, giving a particularly problematic clinical phenotype; these include classical stem-cell pathways such as Wnt/β-catenin (beta), Notch, Hedgehog, and JAK/STAT3 as well as EMT regulators such as SNAIL and TWIST. Moreover, TNBC and other tumors similarly overexpress transcription factors to promote and maintain pluripotency and self-renewal, a common feature of CSCs and normal stem cells; these factors include OCT4, SOX2, NANOG, KLF4, and MYC [24,25]. In recent years, different surface markers were subsequently identified in CSCs. The various combinations used to identify breast cancer stem cells (BCSCs) include CD44^+^/CD24^−/low^, aldehyde dehydrogenase (ALDH1), CD44^+^/CD49f^+^/CD133^+^, and epithelial cell adhesion molecule (EpCAM/CD326), all cell subpopulations that have been shown to have increased tumorigenic and metastatic potential [26,27] as well as also be involved in EMT. In fact, a part of these CSCs undergo important changes during EMT that carry out a shift from a stationary state into a migratory one to become circulating tumor cells (CTCs) [28]. 

MCL1 is an antiapoptotic BCL2 protein family member which regulates the intrinsic mitochondrial apoptotic pathway and interacts with the pro-apoptotic BCL2 family members protecting cells from apoptosis, including the BH3-only sensor proteins such as BIM, NOXA, and PUMA as well as the apoptotic effectors BAX and BAK, which trigger the intrinsic apoptotic pathway following cellular stresses and cytotoxicity [29,30]. MCL1 is an essential protein for the survival of numerous cell types, as demonstrated through the embryonic lethality of germline MCL1 deletion in mice. It is understood that MCL1 is often upregulated in a wide range of solid tumor types, including BC and TNBC [31]. Beyond its role in apoptosis, MCL1 also plays a critical role in the regulation of several cellular processes including embryonic development, mitochondrial homeostasis and bioenergetics, cell cycle control, autophagy, senescence, and invasion [29,32,33]. High expression of the MCL1 protein was considerably associated with poor prognosis, malignant cell growth, and evasion of apoptosis, as well as increased invasive grade. MCL1 is also one of the key regulators of self-renewal in CSCs, conferring a great advantage for tumorigenesis and drug resistance [31]. For this reason, due to its critical role in different pathological settings, research has focused its attention on the discovery and clinical development of selective MCL1 inhibitors able to pharmacologically modulate its functions to target the BH3 binding domain and modulate interactions with other BCL2 members. Among these compounds, the latest and promising A-1210477 possess high selectivity and affinity for MCL1. Nevertheless, because of the complexity of MCL1 regulation and function, the clinical application of MCL1 inhibitors is challenging [32,33]. 

In recent years, our research aimed to understand the molecular mechanism underlying anticancer drug resistance in TNBC tissues and cell lines [34,35,36] as well as in mammospheres and three-dimensional (3D) cultures originated from these cells [37,38]. Currently, our objectives are focused on MCL1 protein, which also plays a key role in self-renewal in CSCs [29]. In this paper, we investigated the effects of MCL1 inhibition using A-1210477, a selective MCL1 inhibitor [39], in MDA-MB-231, a TNBC cell line which shows marked aggressive and invasive features as well as anoikis resistance, EMT characteristics, and a high stemness component [40]. Here, we investigate the molecular mechanisms through which MCL1 inhibition may lead to a reduction in MDA-MB-231’s intrinsic features, analyzing the expression of genes and protein levels related to anoikis/apoptosis, migration, invasiveness, EMT, and stemness. In addition, we demonstrate the direct interaction between MCL1 and proteins involved in focal adhesion (FA) dynamics, proposing that MCL1 also contributes to the maintenance of FA integrity. Altogether, our findings highlight that MCL1 inhibition sensitizes the MDA-MB-231 cell line to anoikis, reducing its main aggressiveness features and, thus, suggesting MCL1 as a putative target candidate for TNBC therapy. 

## 2. Results

### 2.1. Effects of A-1210477 on MDA-MD-231 Cell Viability and Morphology

This study aimed to investigate the role of MCL1 in TNBC, reporting the effects of A-1210477, a BH3-mimetic compound that selectively binds MCL1, on the MDA-MB-231 cell line. First, cell viability was evaluated via MTT assay. As shown in Figure 1A, after 24 h, A-1210477 treatments (5–7.5 and 10 μM) did not affect cell viability, whereas 15 μM reduced the cell viability by 25%. Prolonging the time of treatment to 48–72 h, we observed a progressive dose-dependent cell viability reduction up to reaching IC 50 with 10 μM at 72 h. Phase-contrast microscopy images (Figure 1B) show cellular morphological changes resulting from treatments with respect to untreated control cells. As we can observe, prolonged incubation (48–72 h) with higher concentrations of A-1210477 (10 and 15 μM) induced cell rounding and detachment from the substrate. Thus, for all subsequent experiments, we chose to use the concentration 10 μM between the 48–72 h time range. In addition, a viability assay was also performed in serum-free medium, and surprisingly, we observed faster and drastic effects (about 90% viability reduction) after 1–2 h of treatment compared to serum-free control cells (Appendix A). According to Soutar et al., further analysis identified that bovine serum albumin (BSA) is the protective factor of these drastic and fast events [41]. In fact, the addition of BSA to the culture medium (2.5 mg/mL) restored the initial range of time.

### 2.2. A-1210477 Treatment Induces Anoikis in MDA-MB-231 Cells

As observed via microscopy analysis, A-1210477 treatment leads to rounding, cell–cell and cell–matrix detachment with subsequent cell death. This is a form of programmed cell death known as anoikis [42]. Here, we report the evaluation of some key proteins of anoikis via Western blotting analysis at 72 h of treatment. Figure 2A shows increased BIM levels and contextually decreased BID and c-FLIP levels in treated cells with respect to control cells. Moreover, it is well known that anoikis is a form of cell death which involves both intrinsic and extrinsic apoptotic pathways [9]. Therefore, we also analyzed the characteristic proteins involved in both apoptotic process, i.e., BAX and Bcl2, which are pro- and anti-apoptotic members of the Bcl2 family, respectively [43,44]; Caspase-8 and Caspase-3, master regulators in apoptosis; and their cleaved substrates, PARP and Lamin-B. Figure 2A shows increased levels of BAX as well as decreased Bcl2 levels. In addition, in cells treated with A-1210477, we observed a decrease in Caspase-8 and Caspase-3 levels with the simultaneous appearance of cleaved 43/41 kDa and 19 kDa fragments, respectively, as well as the cleavage of PARP and Lamin-B. Apoptosis was also confirmed via Annexin V flow cytometry analysis (Appendix A). Moreover, Hoechst staining fluorescence microscopy showed apoptotic morphology with chromatin condensation and fragmentation following A-1210477 treatment (Figure 2B).

### 2.3. MCL1 Inhibition Promotes Integrin Switch along with Focal Adhesions (FAs) and Survival Pathway Interruption

Following the observation that MCL1 inhibition induces anoikis, we also evaluated integrin switching, another characteristic event of this process [45]. Western blotting analyses reported in Figure 3A show that A-1210477 treatment induces changes in integrin subunits’ expression., in particular, an increase in αν and a relative decrease in β1 and β3 subunits in treated cells with respect to control cells. Moreover, we evaluated the proteins involved in FA dynamics, such as FAK, p130Cas, and γPak [46,47], but also the downstream signaling AKT/ERK survival pathways [48]. Interestingly, in Figure 3B, we can observe a drastic decrease in FAK levels as well as a reduction in its molecular scaffold p130Cas and γPak, suggesting focal adhesion disruption as a possible consequent event of the integrin switch. Moreover, Figure 3C shows a reduction in pAKT and pERK levels in treated cells with respect to control cells, indicating a survival signaling collapse.

### 2.4. MCL1 Inhibition Reduces Migration and Invasiveness in MDA-MB-231 Cells

It is known that AKT/ERK pathways are involved in several processes such as survival, proliferation, migration, and invasion. As described here, A-1210477 treatment induces the inhibition of these signaling pathways. Therefore, we looked into the possible effects induced via MCL1 inhibition on migration and invasiveness capabilities. Phase contrast images showing a wound healing assay (Figure 4A) confirm that, at 24 h and 48 h after scratching, treated cells migrate more slowly (about 22% and 25%, respectively) than untreated cells. Interestingly, at 48 h, the scratched area is fully closed in control cells. 

Moreover, we performed a Matrigel Transwell invasion assay to evaluate the effects on invasive activity. As can be observed (Figure 4B), treated cells show a reduction in invasive activity (about 45%) with respect to untreated control cells. To confirm these observations, we analyzed some proteins involved in the invasion process such as metalloproteases MMP-2, MMP-9, MT-MMP-1, as well as their inhibitor TIMP-2 [49,50]. As reported in Figure 4C, metalloproteases levels are decreased through A-1210477 treatment, while TIMP-2 levels are increased. These results allowed us to hypothesize that MCL1 can play a key role in the migration and invasiveness abilities of MDA-MB-231 cells.

### 2.5. MCL1 Inhibition Reduces Stemness Features in MDA-MB-231 Cells

MDA-MB-231 cells are also characterized by features associated with CSCs [40]. Therefore, we evaluated if MCL1 inhibition was able to reduce stemness potency in this cell line. *SOX2*, *NANOG*, *OCT3/4*, as well as a CD44^+^/CD24^−/low^ phenotype were reported to be the principal markers associated with stemness in breast cancer [51,52]. The cytofluorimetric analysis of A-1210477-treated MDA-MB-231 cells reported in Figure 5A shows an interesting moderate reduction in the percentage of CD44^+^ as well as an increase in CD24^+^ cells. Contextually, treatment with A-1210477 reduced the percentage of CD326^+^ and CD49f^+^ cells, two other markers of BC stemness [53,54]. In addition, qRT-PCR analysis for *SOX2*, *NANOG,* and *OCT3/4* showed a drastic reduction in mRNA level expression (Figure 5B). Moreover, CD133, another marker involved in stemness phenotype [55] was also evaluated. Figure 5C shows the reduction in its levels. All together, these data suggest that MCL1 inhibition leads to a reduction in cancer stemness in this cellular model.

### 2.6. MCL1 Inhibition Promotes a Possible Reversion of Epithelial–Mesenchymal Transition (EMT)

The epithelial–mesenchymal transition (EMT) is a critical process of tumor cell morphological transformation involved in cancer progression and metastatic potential. The MDA-MB-231 cell line exhibits an enrichment for markers associated with EMT [56]. For this reason, we assessed the effects of MCL1 inhibition on the expression of EMT markers such as E-Cadherin, N-Cadherin, Vimentin, and MUC1 [57,58,59]. Western analyses are reported in Figure 6A. As we can observe, A-1210477 treatment induces N-cadherin, Vimentin, and MUC1 reduction and a marked re-expression of E-cadherin, which is confirmed via qRT-PCR showing an increase in its mRNA levels (Figure 6B). Furthermore, we also evaluated the expression of Cytokeratin 19, another marker downregulated in EMT [60]. Western blotting analysis in Figure 6A shows an increase in this marker following A-1210477 treatment.

### 2.7. The De Novo Production of E-Cadherin via GSK3β Activity and DNMT1 Degradation

It is known that E-Cadherin is poorly represented in the MDA-MB-231 cell line [61]. In order to correlate MCL1 inhibition to de novo expression of E-cadherin, we tried to understand the molecular mechanism implicated in this event. Several studies report that repression of the *E-Cadherin* transcription gene is due to its promoter methylation by DNMTs [62]. Here, we report the DNMT1 expression analysis. As shown in Figure 7A, expression levels of DNMT1 protein decrease, while we did not observe significant changes in its mRNA expression following A-1210477 treatment, suggesting possible protein degradation. As reported, post-translational modifications, such as phosphorylation, regulate the activity and stability of DNMT1, including AKT/GSK3β-mediated phosphorylation [63]. According to decreased pAKT levels (Figure 3C), we demonstrated that A-1210477 downregulated the phosphorylated and inactive GSK3β form (Figure 7A), thus suggesting that enhanced GSK3β activity can lead to phosphorylation and subsequent DNMT1 degradation.

### 2.8. MCL1 Inhibition Promotes Itself Downregulation

NF-κB and STAT3, due to their related pathways, are the main transcription factors involved in the regulation of several cellular process including anoikis resistance, EMT, and stemness [64]. Moreover, they are also the master regulators of the MCL1 expression gene [29]. Therefore, we evaluated and reported the reduction in NF-κB and STAT3 phosphorylation levels in treated cells with respect to control cells (Figure 8A). In addition, this evidence could explain the reduction in MCL1 mRNA levels as confirmed via qRT-PCR analysis reported in Figure 8B. These results suggest that MCL1 inhibition by A-1210477 not only leads to its activity reduction but also to downregulation, affecting main and crucial cellular pathways.

### 2.9. A Possible Non-Canonical Localization and Role of MCL1 in FAs

It is known that MCL1 exerts its function in OMM through sequestering pro-apoptotic factors such as BIM, BID, NOXA, and PUMA [29]. Here, we demonstrated that its inhibition by A-1210477 induces dramatic changes in cell signaling, aggressiveness, and metastatic behavior in MDA-MB-231 cells. So, we assumed a non-canonical role of MCL1 due to its different cellular localization such as on the cell membrane and/or FAs. To demonstrate this, we investigated MCL1/FAK and MCL1/p130Cas interaction through immunoprecipitation analysis. As shown in Figure 9, MCL1 interacts with both proteins, and its inhibition with A-1210477 determines their dissociation. Notably, A-1210477 linked to MCL1 protein promotes its inactivation and accumulation [38], as observed in both MCL1 IP and input. 

## 3. Discussion

Among BC subtypes, TNBC is the most aggressive form with elevated metastatic potential, poor clinical prognosis, and limited treatment options. For this reason, there is a constant need to find new molecular targets in TNBC that could lead to the development of new treatment approaches. Although it is highly auspicious to identify cancer-cell-specific drugs, nowadays research is still far from achieving this goal. 

MCL1 is a crucial antiapoptotic factor involved in the survival of numerous cell types and is often deregulated in several solid tumors, including TNBC. High expression of MCL1 is associated with increased invasiveness and cell growth and plays a role in stemness and drug resistance [31]. In addition, MCL1 is involved in anoikis resistance, including in lung carcinoma cells [65] and cutaneous melanoma [66]. Moreover, MCL1 protein was involved in intrinsic drug resistance in cancer therapy, such as poor response to paclitaxel treatment in a cohort of invasive BC patients which showed increased MCL1 protein levels after treatment [31].

The aim of this study was to evaluate the effects of MCL1 inhibition in MDA-MB-231 cells, an aggressive TNBC cell line with high migratory and invasiveness capabilities as well as stemness, epithelial–mesenchymal transition (EMT), and anoikis resistance features. Here, we inhibited MCL1 in MDA-MB-231 cells using A-1210477, a selective BH3-mimetic.

First, we observed a reduction in cell viability in a dose- and time-dependent manner, obtaining IC50 with 10 µM concentration at around 72 h of treatment and choosing to use this concentration in our experimental set. Meanwhile, the changes in cellular morphology led us to consider that cell death was induced following rounding and cellular detachment, features compatible with a process known as “anoikis”, a form of apoptosis characterized by cell detachment from the extracellular matrix (ECM) or loss of cell–cell contacts [9]. 

Anoikis constitutes a barrier to metastasis; however, some forms of cancer show apoptosis as well as anoikis resistance. Most aggressive BCs, including TNBC cells, often gain anoikis resistance moving from primary sites and metastasizing to distant organs [67]. In addition, D’Amato et al. demonstrated that TNBC exhibits higher rates of metastasis after diagnosis and relatively more resistance to anoikis with respect to ER-positive BC. However, the mechanisms involved are not fully clear, identifying a molecular pathway critical for anchorage-independent cell survival in an NF-κB-dependent manner [68].

Anoikis can be demonstrated through evaluation of several molecular markers such as cellular-FLICE inhibitory protein (c-FLIP); BIM and BID, two proapoptotic BCL2 proteins; as well as several apoptotic markers including BAX and BCL2 and Caspase activation. BIM and BID are activated by cell detachment facilitating the oligomerization of BAX/BAK within the outer mitochondrial membrane (OMM) [9,45]. Our results demonstrated that 10 µM A-1210477 induced an increase in BIM levels due in part to its accumulation following cell detachment and its release from the dynein cytoskeleton [69] but also following release from the MCL1 complex once bound to A-1210477 [70,71]. Contextually, we observed the Caspase-8-dependent BID cleavage and the appearance of tBID fragments, and a decreased c-FLIP level. As it is known, anoikis is a form of apoptotic death; thus, we also evaluated the main apoptotic players, observing an increase in proapoptotic BAX and a relative decrease in antiapoptotic BCL2 levels. Meanwhile, we reported Caspase-8 and Caspase-3 activation as demonstrated through the appearance of their active fragments as well as their elective substrate cleavage from PARP and Lamin B. Furthermore, Annexin V and Hoechst staining confirmed apoptotic events. 

Anoikis is also characterized by integrin switching. Integrins are transmembrane receptors which transduce several survival signals from the ECM into the cell, partly involving FAs, and highlight the great importance of ECM adhesion for cell survival [72,73]. Particularly, is known that β1 and β3 subunit overexpression as well as an increase in ανβ1 and ανβ3 enhance aggressive behavior and metastatic potential in breast cancer, including TNBC [74,75,76]. MCL1 inhibition by 10 µM A-1210477 markedly reduced β1 and β3 subunit expression while the levels of αν were highly increased, a data point that is particularly interesting. In fact, it was reported that integrin αν has some peculiarities with respect to other integrins because it can dimerize with five different β subunits, forming heterodimers that recognize different ECM ligands and appearing as an important target for the development of therapeutic cancer strategies [77]. 

The involvement of integrins in FA signaling pathways led us to investigate the status of some kinases, such as focal adhesion kinase (FAK) and p21-activated protein kinase gamma-PAK (γPAK) as well as its related scaffold protein p130Cas. Interestingly, we observed a drastic decrease in FAK and p130Cas and a moderate reduction in γPAK, suggesting a consistent disruption of FAs and their signaling dropout, presumably following caspase activation [78]. FAK plays a key role not only in FA complexes, where it controls several survival activities, but also in the acquisition of EMT, cell adhesion, migration, and invasion, as well as the maintenance of stemness features. Its autophosphorylation (Y397) induces Src phosphorylation of the FAK catalytic domain at Y576/577 and consequently leads to JAK/STAT3, PI3K/AKT and MEK/ERK signaling. These are survival signaling pathways that can be potential therapeutic targets in TNBC [79]. 

FA signaling interruption affects downstream survival kinase signaling including PI3K/AKT and MEK/ERK pathways. To confirm this aspect, we observed a reduction in both AKT and ERK phosphorylated forms following MCL1 inhibition. AKT and ERK pathways control a plethora of cellular processes such as growth, proliferation, survival, mobility, and invasion [80]. To evaluate the effects of A-1210477 on MDA-MB-231 aggressiveness, we performed functional tests of scratch wound healing and a Matrigel^TM^ assay for migration and invasiveness capabilities, respectively. A 10 µM A-1210477 treatment markedly reduced migratory capability in particular; the wound closure was reduced about 75% after 24 h or 48 h with respect to the untreated cells, suggesting a great reduction in migratory rate. In addition, the Matrigel assay showed a decrease in the number of invasive cells following MCL1 inhibition. Supporting these observations, metastatic markers involved in invasiveness such as MMP2, MMP9, and MT-MMP-1 [81,82] decreased their expression levels; moreover, we observed a simultaneous increase in TIMP2, a promiscuous negative MMP regulator ubiquitously expressed in normal tissues and showing high affinity with pro-MMP2 [83]. These data suggest that both migratory and invasive capabilities of MDA-MB-231 cells can be reduced through MCL1 inhibition.

An important role in the initiation and progression of cancer is sustained by cancer stem cells (CSCs), a subpopulation which contributes to tumor heterogeneity displaying anoikis and drug resistance. Anoikis resistance is also maintained by the tumor microenvironment which enhances CSCs’ self-renewal ability via ECM modulation [84]. In this article, to better understand if MCL1 inhibition is also related to CSCs, we also analyzed the expression of the main surface stemness markers in MDA-MB-231 following A-1210477 treatment. Flow cytometry analyses performed to identify the CD44^+^/CD24^−/low^ population provided evidence that A-1210477 treatment enriched the CD24^+^ subpopulation, lowering that of CD44^+^. This effect was also accompanied by a significant decrease in CD49f^+^ and CD326^+^ cells and some stemness marker genes, namely, *SOX2*, *NANOG*, and *OCT3/4*. Finally, reduced expression levels of CD133 protein are also reported. These data seem to confirm the drastic effects of MCL1 inhibition, even on stemness features, in this cell line model. 

Several studies demonstrated the direct correlation between anoikis, stemness, and EMT in BC, including TNBC [85]. EMT cancer cells acquire invasive and migratory properties through detaching from the ECM or adjacent cells, evading anoikis, and leaving the primary tumor to reach other distant sites. EMT is characterized by decreased expression of cell–cell adhesion molecules such as E-Cadherin as well as increased expression of mesenchymal markers including N-Cadherin [57], MUC1, and Vimentin. Moreover, important transcription factors involved in EMT are also involved in stemness process, these factors include SNAIL, TWIST, and ZEB [59,86]. It is known that MDA-MB-231 is a cell line characterized not only by high stemness features [40] but also by mesenchymal-like properties in which E-Cadherin expression is suppressed through methylation of the promoter [61]. Here, we report interesting data about the increase in E-Cadherin protein due to its mRNA re-expression after A-1210477 treatment, as confirmed via qRT-PCR analysis. Contextually, we obtained a modest decrease in N-Cadherin and MUC1 and a marked decrease in Vimentin levels. In addition, we also evaluated another specific marker downregulated in EMT, Cytokeratin 19 [60], which significantly increased after MCL1 inhibition. In according to this result, Alsharif et al. reported that Cytokeratin 19 is required to maintains E-cadherin localization at the cell surface and stabilize cell–cell adhesion in MCF7 cells [87].

Genetic and epigenetic factors, including DNA methyltransferases (DNMTs) and EMT related genes such as E-Cadherin, N-Cadherin, Vimentin, and MUC1, have a critical role in TNBC [88]. Several studies report that DNMT1 is sufficient to modulate *E-Cadherin* expression via its promoter methylation acting on E-Cadherin expression through its direct interaction with the transcriptional repressor SNAIL [62,89]. In accordance with the literature and with our results regarding E-Cadherin re-expression, we also hypothesized the possible effect on DNMT1 expression following MCL1 inhibition. Notably, after A-1210477 treatment, DNMT1 protein levels were markedly reduced, but not its mRNA, as confirmed via qRT-PCR analysis, suggesting possible protein degradation. In light of this, it is reported that DNMT1 proteasomal degradation is also induced through AKT/GSK3β-mediated phosphorylation [63]. Notably, the phosphorylated and inactive form of GSK3β significantly decreased after treatment, suggesting DMNT1 degradation rather than gene downregulation.

Anoikis resistance as well as invasiveness, stemness, and EMT are very complex processes, and nowadays, their aberrant pathways are not fully understood. These pathways include the activation of pleiotropic transcription factors, kinases, epigenetic modulators, as well as the implications of environmental signals. Among the transcriptional factors, NF-κB is one of the most crucial players in the regulation of numerous biological and pathological processes including invasiveness, stemness, and EMT in concert with the kinase’s activity such as AKT, ERK, and GSK3β [90,91]. 

Another transcription factor involved in EMT and CSC regulation in metastatic process is STAT3 [92]. Given their great importance in these processes, we evaluated the active phosphorylated status of NF-κB and STAT3, which notably resulted in being significatively reduced following MCL1 inhibition, suggesting their complex signaling systems were compromised. Interestingly, NF-κB and STAT3 are both the main transcription factors regulating MCL1 gene expression [29]. Considering this, we analyzed the MCL1 mRNA expression level in our experimental conditions, observing that MCL1 mRNA was markedly reduced, according to the reduced activity of its transcriptional factors. It is interesting to note that A-1210477 binds to MCL1, inhibiting its functions and leading to several disturbances in critical survival pathways in the MDA-MB-231 cell line. On the other hand, this signaling pathway interruption also has consequences in its own downregulation, leading cells to to anoikis/apoptosis through enhancing A-1210477 effects. 

Finally, in view of our findings, we hypothesize, other than its canonical antiapoptotic role, the direct involvement of MCL1 in FA signaling pathways. To confirm this consideration, we identified a direct interaction between MCL1/FAK and MCL1/p130Cas via IP analyses, suggesting a novel function of MCL1 in the signal management and integrity of FAs. For instance, in support of this hypothesis, numerous studies report that FAK signaling controls AKT pathway regulating anoikis and EMT processes [93,94], and AKT directly controls MCL1, regulating its stability [95]. Taken together, our results support the idea that A-1210477 treatment induces drastic effects on the main aggressiveness features in the MDA-MB-231 cell line. In particular, MCL1 inhibition is able to remove anoikis resistance, to reduce migratory and invasiveness capabilities, and to minimize metastatic potential due to stemness and EMT-specific marker downregulation. In addition, this evidence seems to be supported by the relevant engagement of critical survival factors such as AKT and GSK3β as well as NF-κB and STAT3, all factors involved in these processes and which are deregulated following A-1012477 treatment.

## 4. Materials and Methods

### 4.1. Cell Culture Conditions and Reagents

An MDA-MB-231 cell line was purchased from Interlab Cell Line Collection, (ICLC, National Institute of Cancer Research, Genoa, Italy). Cells were cultured in DMEM high glucose integrated with 100 U/mL penicillin, 50 µg/mL streptomycin, 10% heat-inactivated fetal bovine serum (Euroclone, Pero, Italy), 2 mM L-glutamine, and 1% of non-essential amino acids (BioWest SAS, Nuaillé, France). Cells were maintained in a humidified atmosphere in a CO_2_ incubator at 37 °C. For all experiments cells were harvested via trypsinization’ centrifugated at 700 g for 7 min; counted; seeded in a Petri dish (1 × 10^6^ cells), 6-well plate (1.5 × 10^5^ cells), or 96-well plate (6 × 10^3^ cells); and then incubated for 24 h to permit complete adhesion. For experimental procedures, cells were treated with A-1210477 (S7790, Selleckchem, Munich, Germany) or vehicle only at the appropriate concentrations for 24–72 h. Stock solution was prepared in DMSO as suggested by the manufacturer and opportunely diluted in culture medium. The final concentration of DMSO never exceeded 0.03%. All reagents were purchased from Sigma-Aldrich (Milan, Italy), except where stated otherwise.

### 4.2. Cell Viability Assay

A colorimetric cell viability assay was performed to evaluate the dose- and time-dependent effects of A-1210477 on MDA-MB-231 cells. Cells were seeded in a 96-well plate at a density of 6 × 10^3^ cells/well. Then, cells were treated with different concentrations (5–15 µM) for 24–72 h. At the end of the incubation time, we added a solution of MTT (3-(4,5-Dimethyl-2-thiazolyl)-2,5-diphenyl-2H-tetrazolium bromide; Sigma-Aldrich) at a concentration of 11 mg/mL in PBS and incubated for 2 h at 37 °C, as previously described [96]. Finally, 100 µL/well of lysis buffer (20% sodium dodecyl sulphate in 50% N,N-dimethylformamide, pH 4.0) was added and incubated for three hours at 37 °C to solubilize formazan. The absorbance was measured at 540 nm (test wavelength) and at 630 nm (reference wavelength) in a spectrometer reader (OPSYS MR, Dynex Technologies, Chantilly, VA, USA). Cell viability was expressed as the absorbance percentage of treated cells with respect to untreated cells. Moreover, nuclear morphology changes in apoptotic cells were evidenced through Hoechst 33342 staining (B2261; Sigma-Aldrich) as previously described [38]. Briefly, cells were stained with Hoechst 33342 (2.5 μg/mL) for 30 min at 37 °C and visualized via fluorescence microscopy using a DAPI filter. Apoptotic cells were evaluated based on condensed chromatin and fragmented nuclei. Images were acquired on a Leica DC300F digital camera using Leica IM50 software (Leica Microsystems, Wetzlar, Germany).

### 4.3. Western Blot and Immunoprecipitation Analysis

Whole cell lysates were obtained via homogenization in cold radio immunoprecipitation assay (RIPA) buffer containing a protease inhibitor cocktail (Sigma-Aldrich, Milan, Italy) for 20 min. Protein concentrations were determined using the Bradford Protein Assay (BioRad, Hercules, CA, USA). Equal amounts of proteins (25 µg) were subjected to sodium dodecyl sulfate polyacrylamide gel electrophoresis (SDS-PAGE) separation and subsequent electroblotting to transfer proteins on nitrocellulose membranes. Then, immunodetection was performed using primary antibodies incubated at 4 °C overnight and, next, with HRP-conjugated secondary antibodies for 1 h at RT. All experiments were conducted in triplicate. For immunoprecipitation (IP) analysis, 400 μg of proteins were incubated with 0.5 μg of antibody against MCL1 for 2 h and then incubated with 40 μL of IP matrix (ExactaCruz^TM^, Santa Cruz Biotechnology, Santa Cruz, CA, USA) overnight at 4 °C under constant stirring. The immunocomplexes were washed with RIPA buffer, heat inactivated, and submitted to Western blot analysis, as described above, particularly for the detection of MCL1, FAK, and p130Cas. Mock-IP without antibodies were reported as negative control. All antibodies used for Western blotting and IP analyses are reported in Appendix A.

### 4.4. Wound Healing Assay

A scratch wound healing assay was performed to evaluate the migratory capacities of MDA-MB-231 cells with or without A-1210477 treatment, as previously described [97]. Briefly, 1 × 10^6^ cells/well were seeded into 6-well plates. After 24 h, the confluent cells were scratched with a 200 μL pipette tip to generate the scratch. Then, the medium was replaced with medium containing 0.1% FBS to minimize cell proliferation, integrated with 2.5 mg/mL BSA as protective serum constituent. Cells were monitored for 48 h using a Leica DM-IRB microscope. Images were acquired on a Leica DC300F digital camera using Leica IM50 software. The wound area closure was determined using ImageJ software v 1.8.0 and reported as the percentage of relative wound closure. 

### 4.5. Transwell Migration Assay

A Transwell migration assay was performed to evaluate the invasive capabilities of MDA-MB-231 cells with or without A-1210477 treatment, as previously described [35]. Briefly, cells were seeded into 6-well plates at a density of 1.5 × 10^5^ cells/well. After 24 h, cells were treated with 10 µM A-1210477 or only vehicle. After 24 h of incubation, cells were harvested via trypsinization, counted, and re-seeded in an invasion chamber system (BD Biosciences, Discovery Labware, Becton-Dickinson, Italy) in serum-free DMEM (1 × 10^5^ cells/well). In the lower well, we placed DMEM supplemented with 10% FBS as a chemoattractant. Plates were incubated at 37 °C with 5% CO_2_ in humidified air for 24 h. No-migratory cells in the upper chamber were then removed with a cotton-tip applicator. Migrated cells on the lower surface were fixed and stained with Crystal Violet according to the manufacturer’s instruction. Then, cells were visualized under a Leica DM-IRB microscope and images were acquired on a Leica DC300F digital camera using Leica IM50 software. The number of cells migrating was determined through counting ten random fields on each membrane using ImageJ software and reported as percentage of relative invasion with respect to control cells.

### 4.6. Flow Cytometry Analysis of Cell Surface Marker Expression 

For the characterization of surface molecule expression, flow cytometry analyses were performed. MDA-MB-231 cells were managed as previously described in cell culture. After 72 h of A-1210477 treatment, cells were harvested via trypsinization and centrifugated at 700× *g* for 7 min; then, they were resuspended in PBS and counted. After centrifugation, cells were resuspended in ice-cold FACS buffer (PBS, 2% BSA) containing specific antibodies (1:100 *v*/*v*) for 30 min at RT in the dark. The anti-human fluorochrome-conjugated monoclonal antibodies used are reported in Appendix A or with isotype-matched control monoclonal antibodies (BD Biosciences Company, Franklin Lakes, NJ, USA). At the end of incubation, cells were washed twice in ice-cold FACS Buffer, and then samples were acquired on a FACSCantoTM II (BD Biosciences) and analyzed using FlowJo v10 software (BD Biosciences). 

### 4.7. qRT-PCR Analysis 

Total RNA was obtained from MDA-MB-231 cells, set to the experimental conditions described above. Cells were harvested using an appropriate volume of QIAzol Lysis Reagent (Qiagen). RNA was extracted using the Direct-zol RNA MiniPrep (R2050, Zymo Research, Euroclone, Pero, MI, Italy), following the manufacturer’s instructions. After spectrophotometric quantification (Nano Drop2000, Thermo Scientific™, Waltham, MA, USA), 800 ng of purified RNA was subjected to reverse transcription (iScript™ cDNA Synthesis kit, Bio-Rad Laboratories, Milan, Italy) to obtain cDNA, according to the manufacturer’s protocol. For the real-time PCR analysis, a SYBR Green Mix (iQ™ SYBR Green Supermix, BioRad) was used with the specific primers (Appendix A). *GAPDH* was used as a housekeeping gene to normalize the expression level of mRNA. Data were calculated using the 2^−ΔΔCt^ method [98] and expressed as fold change.

### 4.8. Statistical Analysis

The experimental data reported in the Results section were expressed as mean ± SD. Statistical analysis was performed with Student’s *t*-test using Microsoft Excel software v 1.0. Values of *p* < 0.05 were considered statistically significant. When not specified, the data are not significant with respect to the related control. All experiments were conducted in triplicate.

## 5. Conclusions

The data discussed above aimed to support the hypothesis of MCL1’s role in the MDA-MB-231 TNBC cell line as a factor directly involved in cellular mechanisms managing survival, invasion, and metastasis processes. Specifically, its role may be connected to the complex survival pathways of cancer cells, such as its involvement in the dynamics of FAs and in downstream kinase signaling. In our experimental setting, we suppose that A-1210477 binds and inhibits MCL1 protein, promoting its exclusion from the FAs complex and the interruption of survival signaling (Figure 10). The purpose of these studies is to identify MCL1 as an orchestrator of aggressiveness characteristics in MDA-MB-231 cells, one of the most representative cell line models of TNBC. Nowadays, due to absence of specific targets and the need to better understand some mechanisms involved in this pathology, a therapeutic approach in TNBC is very difficult. For this reason, research aims to identify new and promising therapeutic targets to fight TNBC. Considering our results, MCL1 could be proposed for this role.

## Figures and Tables

**Figure 1 ijms-24-11149-f001:**
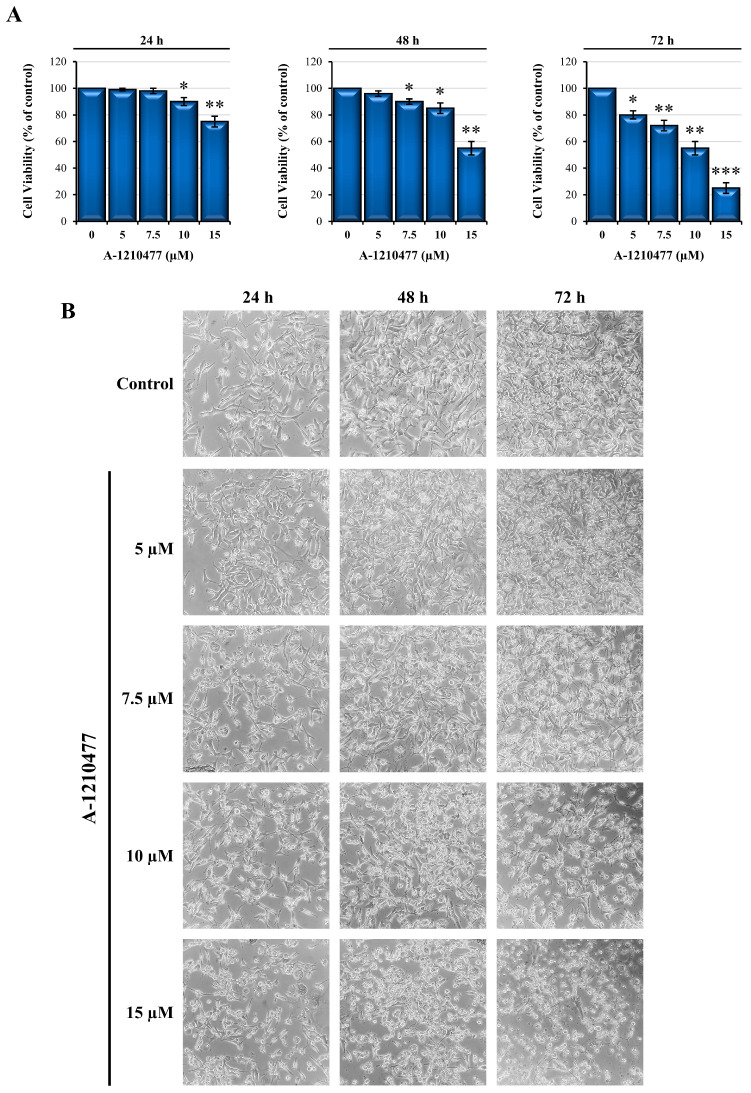
A-1210477 reduces cell viability of MDA-MB-231 cells in a dose- and time-dependent manner. (**A**) MDA-MB-231 cells were treated with increasing doses of A-1210477 (5–7.5–10–15 µM) for different times of incubation. Then, cell viability was analyzed via MTT assay and expressed as the percentage with respect to control cells. (**B**) Phase-contrast microscopy images of MDA-MB-231 cells showing the effects of A-1210477 treatment on cell morphology (magnification 200×). All the experiments were carried out in triplicate. Data are expressed as mean ± SD. * *p* < 0.05; ** *p* < 0.01; *** *p* < 0.001.

**Figure 2 ijms-24-11149-f002:**
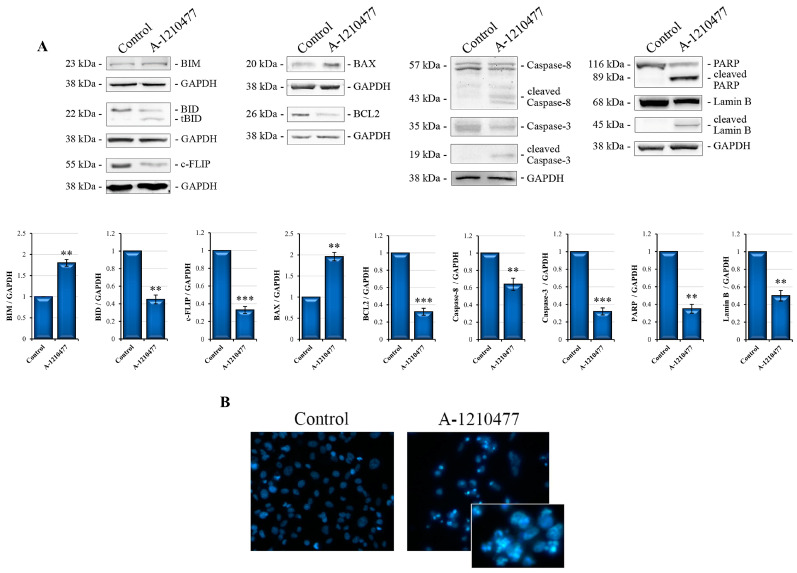
Cell death effects induced by A-1210477 in MDA-MB-231 cells. (**A**) Immunoblot images show anoikis and apoptosis markers. After treatment for 72 h with 10 µM A-1210477, 20 µg of whole cell lysates were submitted to Western blot analyses, using specific antibodies as described in the Section 4. GAPDH was used as a loading control. Images are representative of at least three independent experiments. Data are expressed as mean ± SD in fold changes and are reported in the histogram. ** *p* < 0.01 and *** *p* < 0.001. (**B**) Fluorescence microscopy images of cells stained with Hoechst 33342 dye evidencing chromatin condensation and fragmentation (magnification 200×).

**Figure 3 ijms-24-11149-f003:**
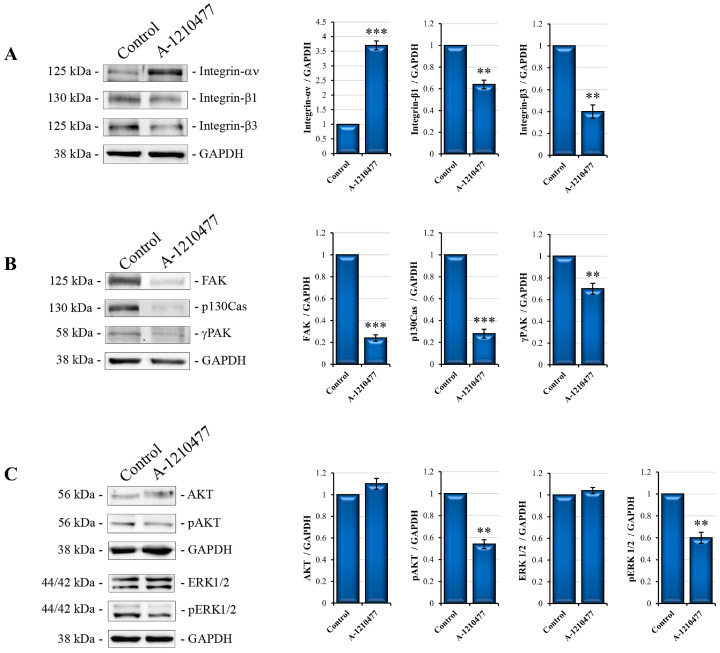
A-1210477 treatment induces changes in integrin expression, FA destruction, and kinase signaling interruption. Western blot analyses show the change in levels of peculiar subunits’ integrins involved in anoikis: (**A**) FA proteins (**B**) and survival kinases AKT and ERK (**C**). All the effects are reached after 72 h of treatment with 10 µM A-1210477. GAPDH was used as an internal control. Images are representative of at least three independent experiments. Data are expressed as mean ± SD in fold changes and are reported in the histogram. ** *p* < 0.01; *** *p* < 0.001.

**Figure 4 ijms-24-11149-f004:**
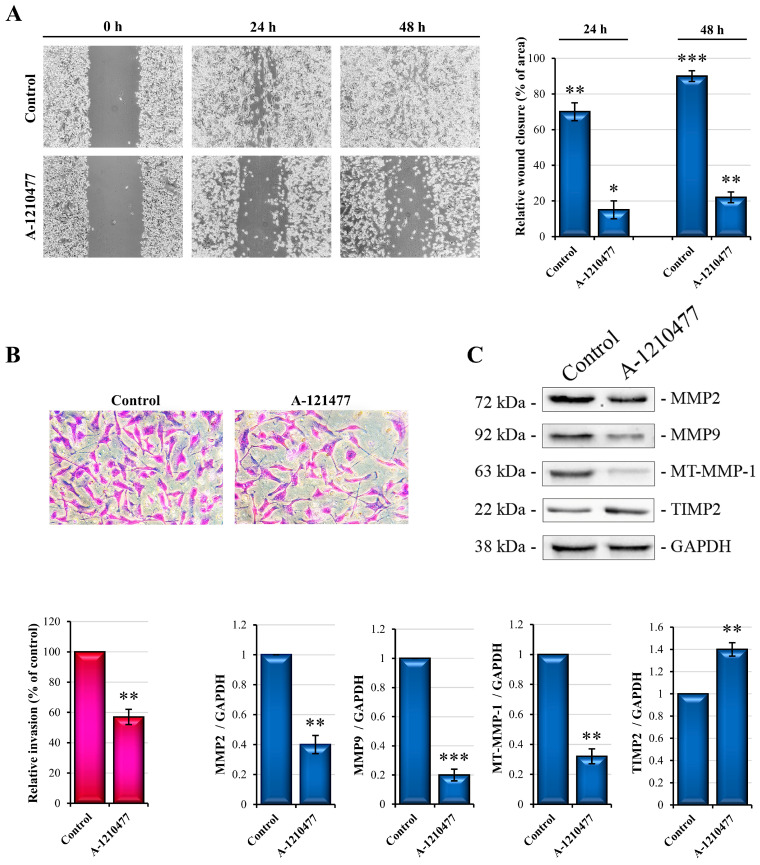
Effects of treatment with A-1210477 on migratory and invasive capabilities of MDA-MB-231 cells. Representative images of phase contrast microscopy reporting scratch wound healing (**A**) and Transwell invasion assay (**B**). After 48 h treatment, cells that migrated to the underside of the insert were stained with Cristal Violet (magnification 200×). Graph summarizes the relative percentage of cells migrating in treated cells with respect to control cells. (**C**) MMP evaluation after A-1210477 treatment. Using specific antibodies, 20 µg of whole cell lysates were submitted to Western blot analyses. GAPDH was used as a loading control. Data represent the mean ± SD. Images are representative of three independent experiments. * *p* < 0.05; ** *p* < 0.01; *** *p* < 0.001.

**Figure 5 ijms-24-11149-f005:**
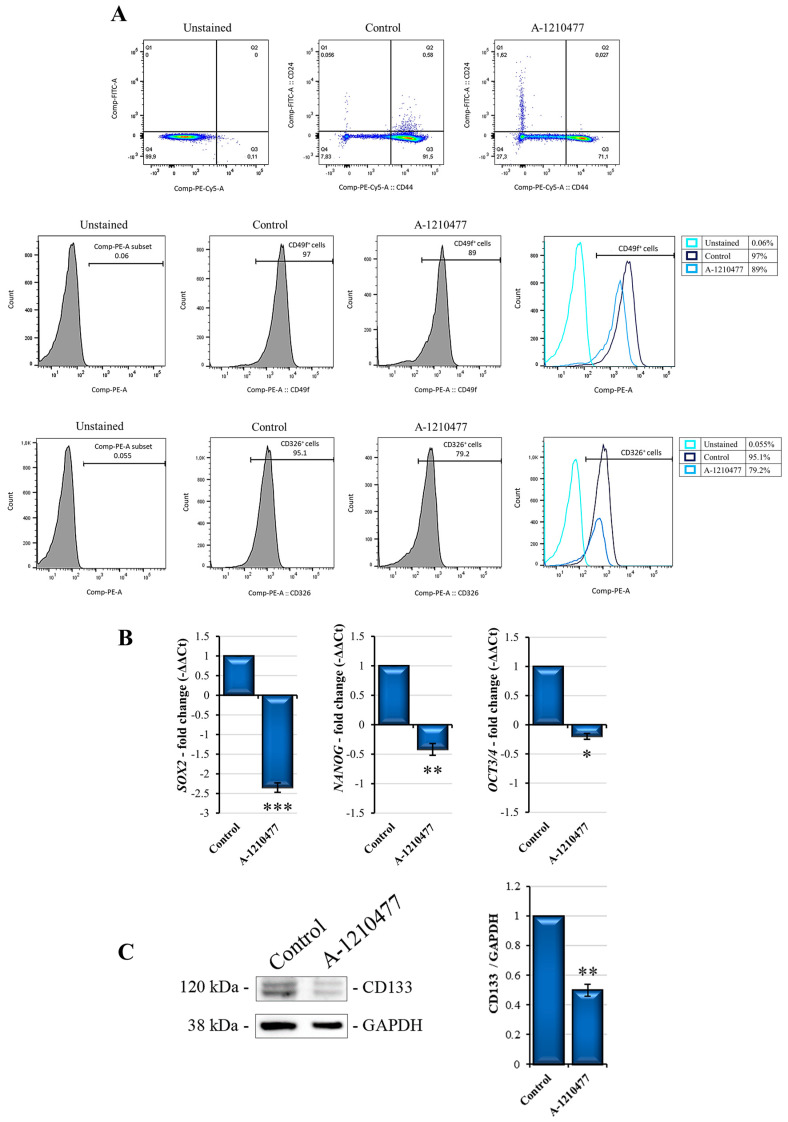
A-1210477 treatment reduces stemness markers in MDA-MB-231 cells. (**A**) Cytofluorimetric analyses show cell-surface expression of stemness markers. After A-1210477 treatment, cells were incubated with appropriate conjugated antibodies (Appendix A) and analyzed via FACS. The histogram plots report the relative expression of CD44^+^/CD24^−/low^, CD49f, and CD326-positive cells in treated cells with respect to the control. (**B**) qRT-PCR analysis of gene stemness-related *SOX2*, *NANOG*, and *OCT3/4*. (**C**) Stemness marker CD133 Western blot analysis. Gene and protein expression was normalized with GAPDH and expressed as fold changes. All the experiments were performed in triplicate. Data are expressed as mean ± SD. * *p* < 0.05; ** *p* < 0.01; *** *p* < 0.001.

**Figure 6 ijms-24-11149-f006:**
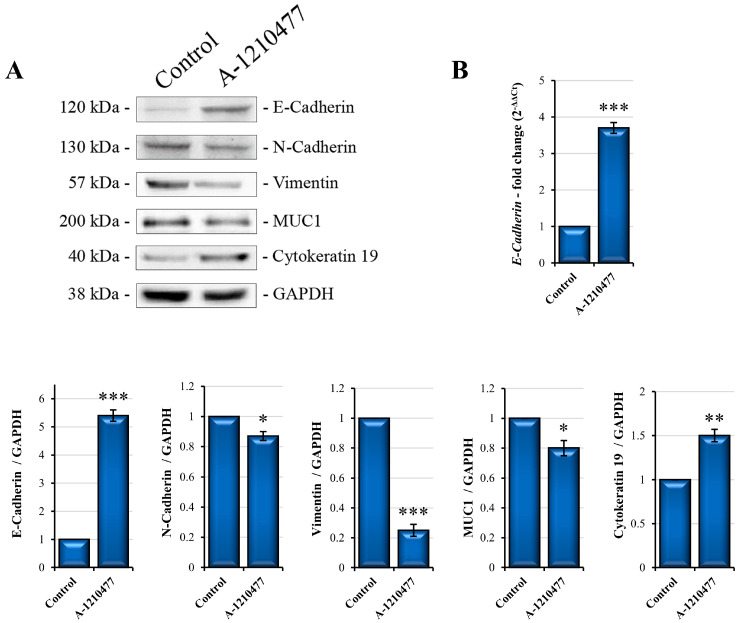
EMT expression markers in MDA-MB-231 cells after A-1210477 treatment. (**A**) E-Cadherin, N-Cadherin, Vimentin, MUC1, and Cytokeratin-19 Western blot analyses in MDA-MB-231 cells treated with A-1210477 with respect to control cells. (**B**) qRT-PCR of *E-Cadherin* in MDA-MB-231. Gene and protein expression was normalized with GAPDH and expressed as fold changes. All the experiments were performed in triplicate. Data are expressed as mean ± SD. * *p* < 0.05; ** *p* < 0.01; *** *p* < 0.001.

**Figure 7 ijms-24-11149-f007:**
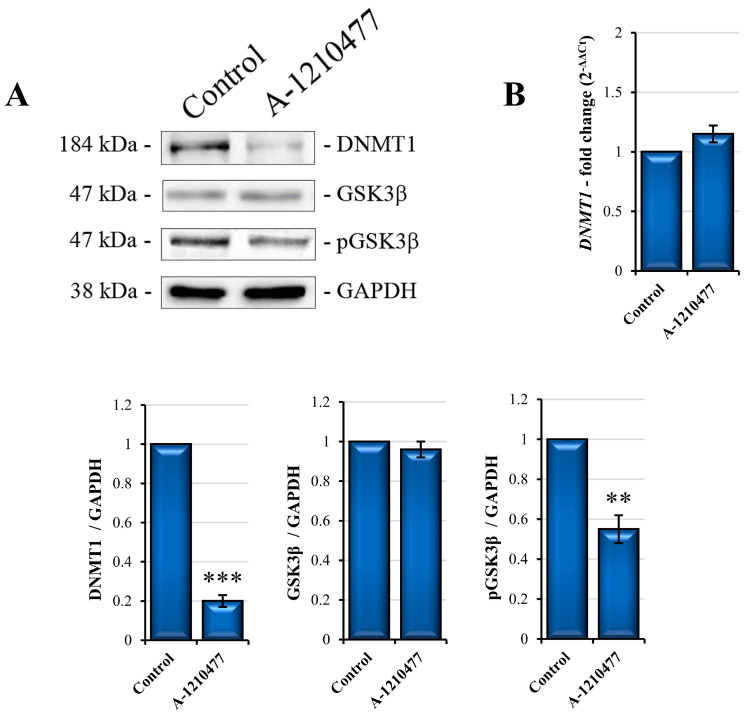
E-Cadherin de novo expression mediated by GSK3β activity and DNMT1 degradation. (**A**) DNMT1 and GSK3β Western blot analysis. The blot image shows a reduction in DNMT1 levels and a decrease in phosphorylated-form GSK3β. (**B**) qRT-PCR of *DNMT1* displays a non-significative change in mRNA expression suggesting protein degradation. Gene and protein expression was normalized with GAPDH and expressed as fold changes. All the experiments were performed in triplicate. Data are expressed as mean ± SD. ** *p* < 0.01; *** *p* < 0.001.

**Figure 8 ijms-24-11149-f008:**
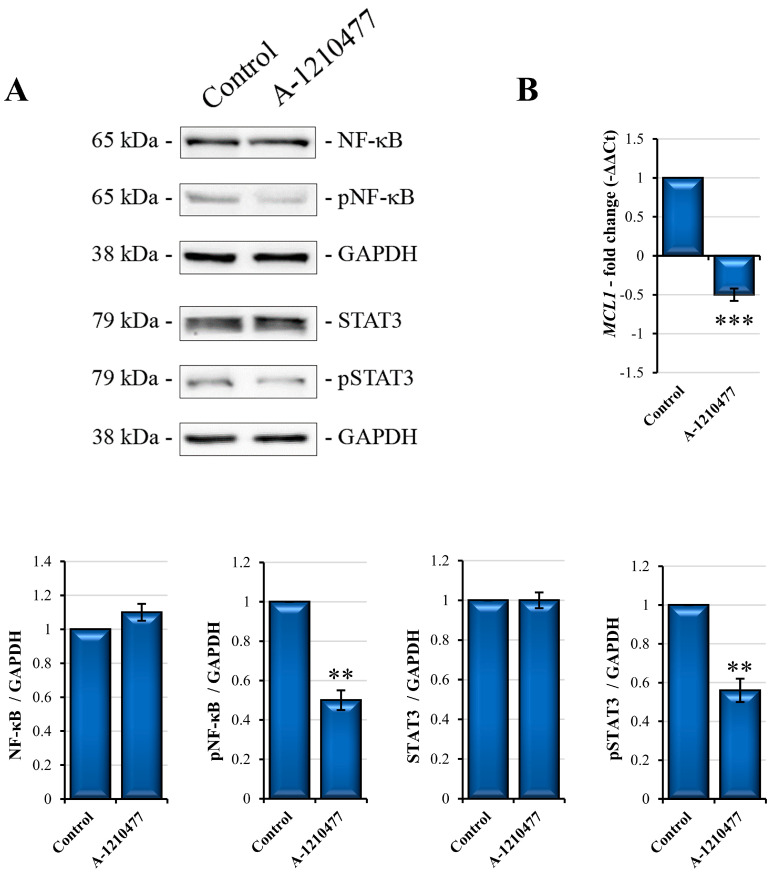
NF-κB and STAT3 phosphorylated forms reduction in MDA-MB-231 cells by A-1210477 treatment promotes MCL1 downregulation. (**A**) Western blot analyses of NF-κB and STAT3 phosphorylation. (**B**) qRT-PCR of *MCL1* expression levels in MDA-MB-231 treated cells with respect to the control cells. Gene and protein expression was normalized with GAPDH and expressed as fold changes. All the experiments were performed in triplicate. Data are expressed as mean ± SD. ** *p* < 0.01; *** *p* < 0.001.

**Figure 9 ijms-24-11149-f009:**
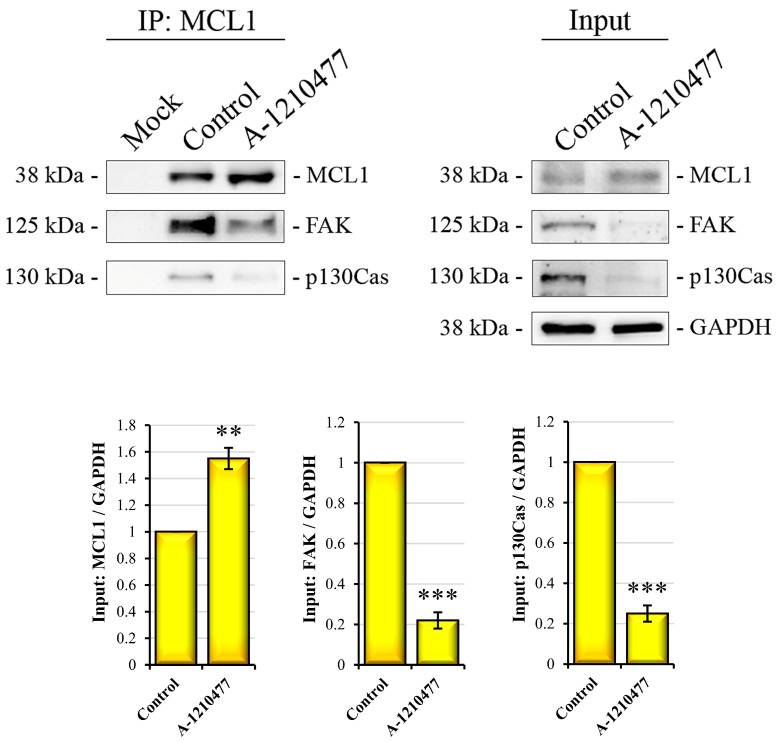
MCL1/FAK and MCL1/p130Cas interaction. Images reporting IP analysis in MDA-MB-231 treated cells with respect to the control cells show the interaction of MCL1 with FAK and p130Cas. IP was performed as described in the Materials and Methods section. In the left panel are Western blot images of the immunoprecipitated proteins (400 µg) with MCL1 antibody and detected with MCL1, FAK, and p130Cas. Mock lanes are obtained by loading beads alone. The right panel report Western blot analysis of MCL1, FAK, and p130Cas performed in input proteins (20 µg). Protein expression in input was normalized with GAPDH and expressed as fold changes. All the experiments were performed in triplicate. Data are expressed as mean ± SD. ** *p* < 0.01; *** *p* < 0.001.

**Figure 10 ijms-24-11149-f010:**
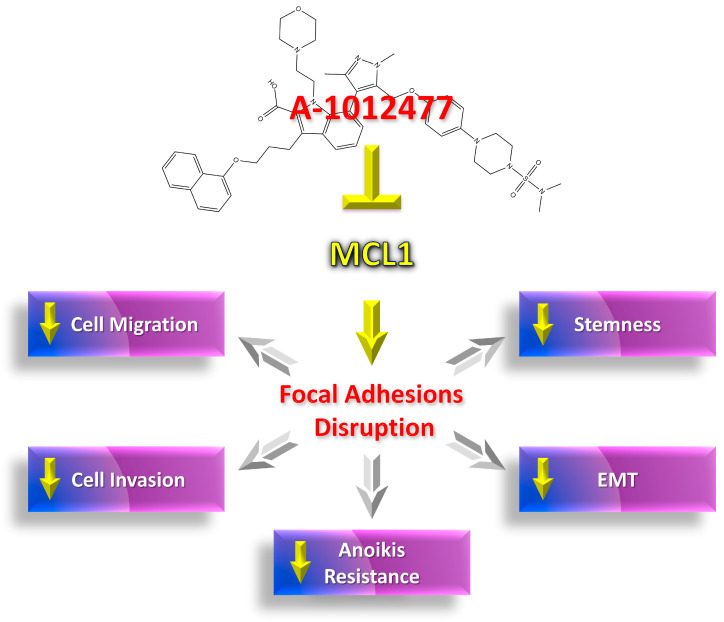
MCL1 Inhibition by A-1210477 reduces the aggression signatures in MDA-MB-231 cells via FA disruption.

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
