# Peer review of "MCL1 Inhibition Overcomes the Aggressiveness Features of Triple-Negative Breast Cancer MDA-MB-231 Cells"

_ijms, 2023, doi:10.3390/ijms241311149_

Round 1

Reviewer 1 Report

This manuscript presents in vitro studies in a triple negative breast cancer cell line treated with an inhibitor of MCL1. Characterization of changes in protein and gene expression levels across several cellular pathways as a result of the inhibitor are presented. This work builds upon prior studies by the investigators using this inhibitor and characterizing signaling pathways in triple negative breast cancer cells. The methods used and results presented are overall consistent with the conclusions discussed by the authors, although it should be noted that the generalizability of these conclusions are limited by the experimental design (only one cell line and only one type of MCL1 inhibition investigated). The following points indicate items for the authors to consider to strengthen the manuscript. 

Points to be addressed:

-Figure 5B and Figure 8B: Please correct the graphs showing the fold-change values for 2-DDCt, because a decrease in expression should still be a positive value (lower than 1, above 0) when plotted on the 2-DDCtscale (negative values could be reported for -DDCt alone, not for 2-DDCt, when expression decreases).

-Figure 6A: Please correct the graph of cytokeratin 19/GAPDH in Figure 6B to match western blot data in 6A and the text in the results (graph indicates a decrease relative to the control, while western blot and text indicate an increase).

Additional comments:

-It is recommended to use a period (.) instead of a comma (,) to indicate decimal places in numbers throughout the manuscript.

-It is recommended to make revisions throughout much of the discussion section (and elsewhere in the manuscript) for proper English grammar and punctuation. 

-Line 158: it seems the authors meant “induced” instead of “not induced”.

-While the authors have provided an extensive amount of background and related information in the introduction and discussion sections, these sections seem very long relative to the scope of the studies presented. Perhaps abbreviating these sections could be considered.

It is recommended to make revisions throughout the manuscript, especially the discussion section, for proper English grammar and punctuation. (For instance, line 247 should read "with respect to", and line 250 should place the comma before "while" instead of after "while". Many more instances of similar corrections occur throughout the manuscript)

Reviewer 2 Report

The author here demonstrated that A-1210477, a MCL1 inhibitor, promoted anoikis and apoptosis in MDA-MB-231 cell line through multiple signaling pathways for programmed cell death. The MCL1 inhibition could further reduce many aggressive features such as migration, invasiveness, EMT and stemness for triple negative breast cancer (TNBC). Overall, this work is well organized and provides enough evidences for the conclusion. I have some constructive suggestions for the further consideration before the final acceptance.

 1) I suggest the author to compare the biomarker of anoikis and apoptosis together in one figure. It will be more impressive to feel which cell death pathway is dominating in A-1210477 treated cells.

2) Besides the cell viability assay, the quantification of cell apoptosis ratio with flow cytometry is highly recommended.

3) The western blotting of MCL1 protein is needed to validate the effect of A-1210477 on MCL1 protein expression, although the qPCR result is provided in Figure 8B.

4) If siRNA is used for the downregulation of MCL1, is there the same effect in MDA-MB-231 cell line as A-1210477?

5) The resolution of some figures should be further improved such as Figure 1B, Figure 2C.

6) A summary of the potential molecular mechanism of MCL1 inhibition on MDA-MB-231 Cells is needed to be supplemented as Figure 10.

English language is fine. No further polishing is needed. 

Round 2

Reviewer 2 Report

The paper is well revised according to the reviewer's suggestions. It can be accepted for publication. It will be stimulating more discussion about the therapeutic benefits of targeting MCL1 with diverse gene regulation tools for more effective cancer therapy.